# COVID-19 Case Tracking System in Quarantine Policy: Focus on the Privacy Shift Concept and Application in South Korea

**DOI:** 10.3390/ijerph191811270

**Published:** 2022-09-07

**Authors:** Kwansik Moon, Nackhwan Kim, Jemin Justin Lee, Hyunsik Yoon, Kyungho Lee

**Affiliations:** 1The National Assembly of the Republic of Korea, Seoul 07233, Korea; 2School of Cybersecurity, Korea University, Seoul 02841, Korea; 3College of Medicine, Korea University, Seoul 02841, Korea; 4Center for Information Security Technology (CIST), Korea University, Seoul 02841, Korea

**Keywords:** COVID-19, privacy shift, South Korea, quarantine policy, pandemic, epidemiological investigation system, cell broadcast system

## Abstract

This study analyzed how Korea’s quarantine policy manages personal information to prevent and control COVID-19. Korea effectively halted the spread of COVID-19 through epidemiological investigations and cell-broadcast systems. In this process, the route of infection is presented without identifying the patient, and the necessary participants are selected only through authentication. We found a correlation between the number of emergency text messages sent by the Ministry of Interior and Safety in 2020 and the number of confirmed cases (R^2^ = 0.465, *p* < 0.001). Based on Korea’s case, we propose a new concept for solving the personal information problems that might arise during a pandemic response.

## 1. Introduction

Despite the short history of the term “privacy,” there has been a continuous shift in its meaning corresponding to social changes. US attorneys Samuel D. Warren and Louis D. Brandeis first recognized the “right to privacy” in 1890 in a case that established the concept of public recognition of private space and realm. Katz v. United States, 389 US 353 (1967), defined the scope of constitutional rights, which allowed privacy to later develop into a fundamental right that required active protection [1]. Since then, the prominence of the internet has gradually expanded to the privacy domain. The protection of personal information through anonymization is recommended worldwide.

Privacy issues associated with the COVID-19 pandemic are legitimate as quarantine authorities’ access to information has risen steadily [2]. Government officials worldwide have expressed concerns over privacy rights guidelines, and policies that test the data collection methods implemented during the pandemic may remain unexamined and unchanged for the next decade [3]. Consequently, a shift in the definition of privacy concepts has contributed to damage reduction during the global pandemic. The South Korean case epitomizes this matter.

Specific responses to the COVID-19 pandemic have been expressed in various ways depending on the country’s governance system, law, and culture [4,5]. Therefore, properly analyzing the COVID-19 response system requires consideration of the nature and specificity of the country’s quarantine policy. A pandemic occurs worldwide, but the response is expressed differently from region to region. This study was not based on pure epidemiological analysis but on derived implications based on policies implemented by the Korean quarantine authorities during the COVID-19 pandemic.

This study sought to demonstrate a privacy-protection mechanism using South Korea’s COVID-19 data. In correlating the number of emergency text messages sent by the Ministry of the Interior and Safety and the number of confirmed cases, it is meaningful to confirm that the protection of personal information was working in Korea during this period. In the past, studies tracked COVID-19 reverse contacts and personal information protection. However, that approach was limited in that it was taken without considering the system change process of the relevant quarantine authorities [6,7]. This study analyzed the government’s epidemiological investigation support system for COVID-19 control and personal information exposure minimization through a cell broadcast system (CBS) by tracking messages for CBS and the number of new COVID-19 cases in South Korea. This study provides new insights for disaster prevention research and related policies.

## 2. Background

In 2015, the Middle East respiratory syndrome (MERS) incident caught the interest of the Korean government and citizens and sparked controversy regarding patient data privacy. At that time, Korea lacked efficient identification methods to prevent the spread of the MERS virus [8]. The outbreak motivated the public and private sectors to re-evaluate personal data management and Koreans’ right to privacy. To overcome the shortcomings of the MERS response, the Korean government established Article 34-2 of the Infectious Disease Prevention Act to allow government agencies to access individuals’ location timelines (infection tracking) [9]. Although infection tracking has helped Korea control COVID-19, data collection involving sensitive information has raised concerns regarding privacy infringement [10].

Korea has demonstrated that mitigating the disease during the initial phase of an outbreak is critical. Currently, the country’s quarantine authorities collect and use detailed personal information about relevant populations—patients, pathogen carriers, and individuals with possible contagion—to monitor the disease status [11]. During this phase, far more sensitive statistics can be gathered, such as individual prescriptions and medical records. These digital interventions could threaten the rights to privacy, equality, and fairness. Article 23 of the Personal Information Protection Act regulates and protects basic privacy rights.

The Korean government introduced an epidemiological investigation system in 2020 to respond quickly to the spread of infectious diseases. This system was prepared through amendments to the National Assembly Law after the 2015 MERS outbreak to enable accurate epidemiological investigations of future health crises. The system minimizes information acquisition according to strict guidelines for safe management of personal information.

## 3. Methodology and Data

To analyze whether Korea’s quarantine policy affected the prevention and control of COVID-19, data related to Korea’s CBS system were collected. Figure 1 shows the system to prevent additional confirmed cases with a detailed explanation based on information on confirmed COVID-19 cases in Korea. Data were analyzed by collecting and identifying the daily number of confirmed cases published by the Korean government and the daily number of emergency disaster messages sent by the Ministry of Public Administration and Security. The study period was from January 2020 to August 2020, when the coronavirus appeared in Korea. The variables were set as the number of new COVID-19 cases and the number of emergency text messages sent by the Ministry of Public Administration and Security. Correlation analysis was performed to analyze the degree of correlation between the two variables. Figure 2 shows the results of analyzing this correlation.

## 4. The Privacy Shift Concept, and Its Application in Korea

Identification and authentication are essential processes in healthcare [12]. However, there must be a balance between protecting public health and individuals’ rights to privacy [13,14]. Therefore, the identification of specific individuals must be avoided unless it is absolutely necessary (e.g., to prevent an infected person from traveling abroad), and authentication should be performed with caution. During a pandemic, failure to identify threats can shake public trust as easily as heavy-handed measures [15]. Maintaining public trust is essential for the smooth functioning of the national governance system in a pandemic situation such as COVID-19 [16].

Therefore, Korea sought the best compromise to effectively control the pandemic, while maintaining privacy by refining identification and authentication. This was a useful part of the quarantine process because it enabled us to collect epidemiological data during the early stages of the COVID-19 outbreak without disclosing individuals’ personal information.

During the lifecycle of personal information, the system requires identification and authentication only during the initial collection stage. In subsequent steps, we separated identification and authentication and performed the required procedures using only authentication information to ensure maximum privacy and safety.

Korea’s quarantine authorities followed the guiding principles regarding personal information. As noted above, we collected only personal information on patients with infectious disease, pathogen carriers, and individuals suspected of contracting an infectious disease. We used the CBS to send COVID-19 information to citizens without identifying the individual recipients. As the system receives only the recipients’ location information, there is no risk of privacy exposure.

In the personal information collection stage, we collected the identification and authentication information. During the usage stage, we removed the identification and pseudonymized authentication information as much as possible. We collected the data used for quarantine through a combination of identification and authentication processes. After the initial identification phase and a confirmed case of COVID-19, we removed the data that identified the individuals. Once the self-quarantine phase ended, we pseudonymized the individual’s data and managed the case only using information that authenticated the individual.

We pseudonymized the authenticated information when performing essential procedures for public services that use personal information. This minimizes the privacy exposure. When government agencies perform essential procedures that require personal information, authenticated information is pseudonymized to protect individuals’ privacy and rights. This minimizes information exposure while enforcing a privacy shift through pseudonymization and inclusion of minimal authentication information.

South Korea’s privacy-shift quarantine concept and model during the COVID-19 pandemic were applied to its overall control system. By shifting the concept of privacy protection from the two core functions of identification and authentication to the areas of authentication and object inclusiveness, we constructed a balanced information system that preserved individual privacy (Figure 1).

## 5. Results

The Korean government implemented two effective quarantine measures in response to the COVID-19 pandemic. The first was a tracking method that used the COVID-19 epidemiological survey support system. When the patients tested positive for COVID-19, the system collected their location and payment history to track the possible path of infection. A comparison between the patients’ domestic and foreign route measurements revealed a significant difference. As of 12 July 2021, 169,146 confirmed cases of COVID-19 had been reported since the initial outbreak. Of these, only 36,206 cases (21.4%) were not confirmed via infection routes.

Korea’s neighbor, Japan, is sensitive to the protection of personal information and has been strongly reluctant to convert hospital medical records into data [17]. It is thus difficult to conduct government-led epidemiological investigations in Japan. Therefore, it is difficult to identify additional potential COVID-19 cases. This is problematic given that Japan had more confirmed COVID-19 cases than Korea until December 2020, and 60.9% of the Japanese cases could not be traced following the initial outbreak. The Korean government uses an epidemiological investigation support system to prevent the spread of the virus through the rapid and efficient verification of transmission routes.

The measured value is trivial compared to that of Japan, which had the largest number of COVID-19 infections, with 60.9% unknown routes from the initial outbreak of COVID-19 until December 2020. Thus, the Korean government’s use of an epidemiological investigation support system to prevent the spread of the virus through rapid and efficient verification of transmission routes is notable.

The second method implemented by Korea is information-sharing using the CBS. Officials present legally collected information as a spatiotemporal risk factor. Based on the region’s probable risk and time associated with the biological evidence, real-time updates were added for efficient delivery. Infection alerts were sent to people in the affected areas and selectively categorized according to various factors, such as the minimum contact distance that would make someone susceptible to viral transmission and incubation periods.

Using these two methods, Korea identified infection sources without personally identifying patients, and the necessary message recipients were selected only through authentication. This new system significantly minimizes privacy infringement and threats. For cases where the exposed identifier was too specific, the range of citizen selection was expanded to achieve a balance between preserving privacy and protecting public health.

Figure 2 shows the correlation between the number of alert messages sent by the Ministry of the Interior and Security, and the number of patients diagnosed in 2020 (R^2^ = 0.465, *p* < 0.001). The difference in the time interval between the fluctuations in the two graphs implies a prompt information disclosure. These emergency alerts can relieve public anxiety by establishing a certain level of confidence in preventing and mitigating pandemics.

The CBS sends emergency text messages related to COVID-19 to selected citizens by setting the base station as the minimum unit. This mechanism does not address the recipient-identification steps; it only receives the location information of the receiver, which naturally limits the risk of privacy exposure. While performing the essential procedures of public services that require personal information, the system pseudonymizes the authenticated information. This minimizes the exposure to sensitive data through a minimum number of authentication protocols. This concept forms the basis for the model applied to South Korea’s overall control system.

The CBS does not reveal any information, and only the information necessary for COVID-19 prevention is promptly delivered to at-risk individuals. This process has a structure similar to that of zero-knowledge proof. It provides maximum anonymity, which Korea uses to establish and maintain an effective quarantine system.

## 6. Conclusions

Local authorities and government agencies in South Korea have mitigated the pandemic while ensuring the privacy rights of citizens. Digital measures help public health experts map, monitor, and alleviate diseases, proving that they remain vigilant about respecting citizens’ rights to privacy. South Korea’s privacy shift concept and model succeeded because it revised the identification and authentication processes to ensure ethical and fair practices during the COVID-19 pandemic.

This study found that separating the identification and authentication processes was the most important factor. We propose that this method be developed as a global standard. Although different countries may face additional challenges, the general concept appears to be an accessible and feasible method applicable at a realistic level. When a new pandemic such as COVID-19 occurs, and the need for privacy protection during the quarantine process will remain high. Therefore, it is necessary to manage new data-driven systems by protecting privacy based on authentication methods and extensions rather than traditional identification methods.

As previously mentioned, pandemics are global events but their responses are always local. The purpose of this study was not to compare Korea’s case with other countries but to highlight a system that could be adapted for other countries in the early stages of the pandemic. Our results were based on Korea’s governance system, laws, and culture and may not be generalizable to other countries without considering their governance system, laws, and culture.

We analyzed Korea’s response system to prevent the spread of infectious diseases during the early stages of COVID-19 by examining the correlation between the government’s emergency text messages and the confirmed cases. We found that providing accurate information helped to prevent the spread of COVID-19 by identifying potential hotspots without exposing personal information.

## Figures and Tables

**Figure 1 ijerph-19-11270-f001:**
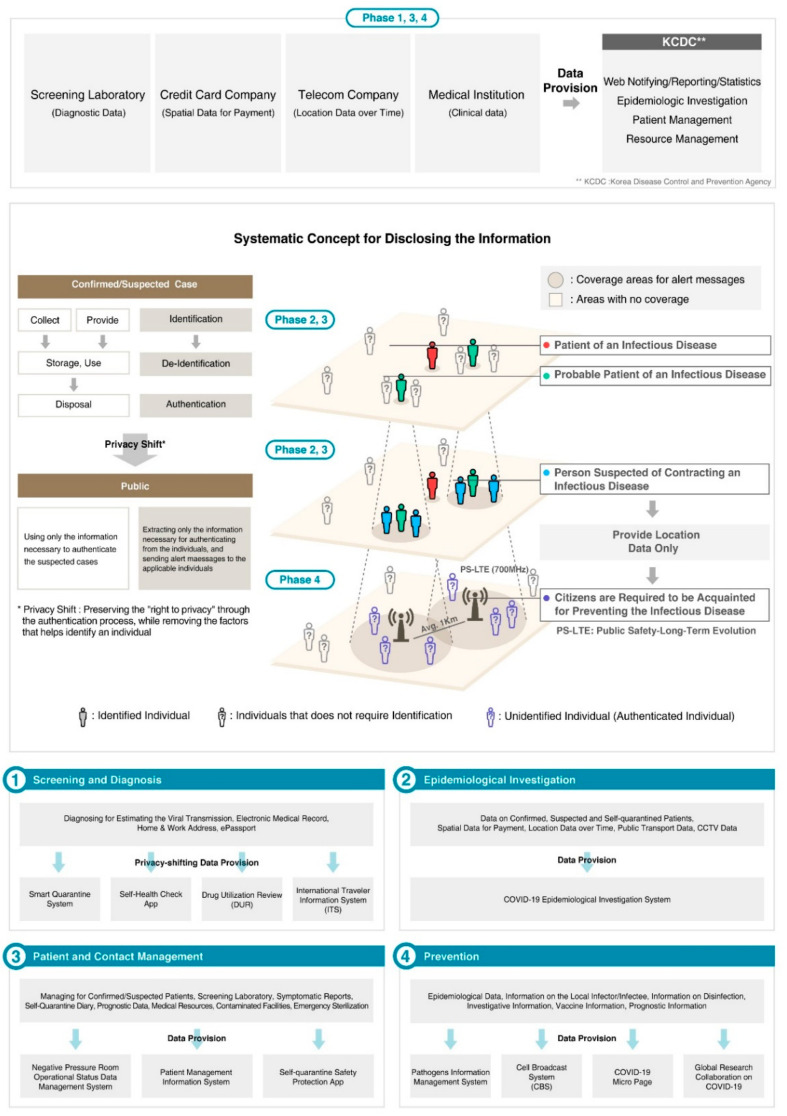
The privacy shift concept and application in South Korea.

**Figure 2 ijerph-19-11270-f002:**
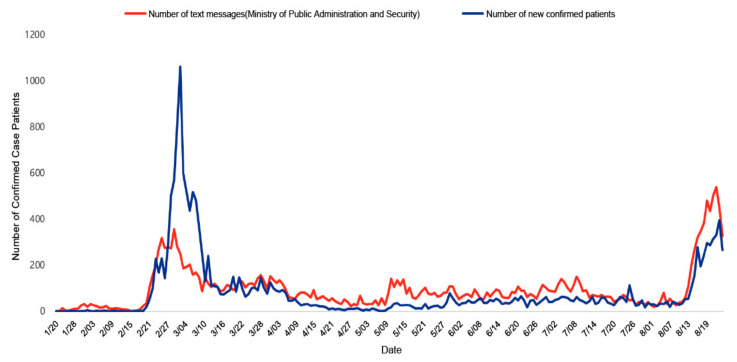
Analysis of the number of new cases of COVID-19 in South Korea in 2020.

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
