# Peer review of "COVID-19 Case Tracking System in Quarantine Policy: Focus on the Privacy Shift Concept and Application in South Korea"

_ijerph, 2022, doi:10.3390/ijerph191811270_

Round 1

Reviewer 1 Report

The study aims to analyze the COVID-19 epidemic situation and explore mitigation methods that examine the timing and control measures of the Korean authorities. The idea is not new but still might be interesting. The manuscript structure is transparent and covers the scope of the Journal. This short report could be published (as a communicate) if major improvements are made:
1.    The keywords should be shortened e.g. “South Korea Quarantine Policy“ into “South Korea”; “Quarantine Policy“
2.    In the Abstract and the Introduction section please stress why the research are important, so the results should be distributed quickly in the form of communicate.
3.    Please indicate shortly the main results in the abstract.
4.    The subjects related to COVID-19 mitigation methods have been investigated over the past three years. Why is this study original? What is new/fresh in the manuscript? Please highlight it in the abstract and stress in the text.
5.    “The Privacy Shift Concept and Application in South Korea” diagram description should be more detailed.
6.    Are similar methods used in other countries, described in the literature? Please include a short analysis.
7.    Does presented method have any drawbacks limitations, if so please indicate them.
8.    In the conclusion section please write how the findings from the study can be applied or utilized.
9.    The references section has only 7 items, it should be developed. Include preferably reports, papers published in 2021 or 2022.

Reviewer 2 Report

1.     The purpose of this paper is unclear. Do you purpose to explain the COVID-19 tracking system in South Korea? Or you aimed to analyze the effects of such system on the prevention and control for the COVID-19 in South Korea? You should clarify in the Introduction.

2.     You should re-structure the manuscript. Section 3 (Analysis of Results) described the results of analysis without any explanations of the analysis the authors performed. Also, I don’t understand why the introduction of the system comes earlier than the results.

3.     You did not clearly explain the study period. For example, which year in Figure 1?

4.     There are many published papers about the association between contact tracking and privacy, but you did not extensively review previous studies.

5.     You should check the academic English. For example, I think “Analysis of Results (Section 3)” should be “Results”. It does not about

Reviewer 3 Report

This paper used COVID data in South Korea to showcase a mechanism of privacy protection. The data presented in Figure 1 is interesting, and the privacy shift concept seemed working in South Korea.

Reviewer 4 Report

The work can be applied when there is an organization which knows the location history of patients.  In such a situation, it provides a practical solution for alerting people with small privacy disclosure.

A minor comment is as follows: the authors argued that the separation of identification and authentication is the most important factor. However, it is hard to find what their definitions are. For example, it can be applied to recipients. But I am not sure it can be applied to patients, too?

Round 2

Reviewer 2 Report

1. The authors need to carefully structure the paragraphs. In general, the first sentence stands for the whole idea of each paragraph. In this sense, the current form of the manuscript does not look like that.

- For example, "Research has been conducted on contract tracking and privacy [5,6] (line 52). This sentence sounds more like the literature review, not about the general introduction of the paper. Given that the last paragraph in the academic paper describes the aims, contents, methods, and implications within short sentences. 

2. There is no section explaining the method. I am still curious about how the potential readers can understand the results without explanations of method.

3. Many newly added arguments do not come with the appropriate references. To persuade the readers (or reviewers), it would be better to say something with the references.

For example, "Identification and authentication are essential processes in healthcare. However, there must be a balance between protecting public health and individual's rights to privacy". 

4. It would be better to have Section 4 earlier than Section 3. Section 4 seems like how the South Korea has succeed in the tracking (or something). Then, Section 3 (Results) can be understood. 
